

# Multi-source SO$_2$ emissions retrievals and consistency of satellite and surface measurements with reported emissions

Vitali Fioletov[1], Chris A. McLinden[1], Shailesh K. Kharol[1], Nickolay A. Krotkov[2], Can Li[2,3], Joanna Joiner[2], Michael D. Moran[1], Robert Vet[1], Antoon J. H. Visschedijk[4], and Hugo A. C. Denier van der Gon[4]

[1] Air Quality Research Division, Environment and Climate Change Canada, Toronto, Canada.
[2] Atmospheric Chemistry and Dynamics Laboratory, NASA Goddard Space Flight Center, Greenbelt, MD, USA.
[3] Earth System Science Interdisciplinary Center, University of Maryland College Park, MD, USA
[4] TNO, Department of Climate, Air and Sustainability, Utrecht, the Netherlands

*Correspondence to*: Vitali Fioletov (Vitali.Fioletov@outlook.com or Vitali.Fioletov@canada.ca)

**Abstract**. Reported sulfur dioxide (SO$_2$) emissions from U.S. and Canadian sources have declined dramatically since the 1990s as a result of emissions control measures. Observations from the Ozone Monitoring Instrument (OMI) on NASA's Aura satellite and ground-based in-situ measurements are examined to verify whether the observed changes from SO$_2$ abundance measurements are quantitatively consistent with the reported changes in emissions. To make this connection, a new method to link SO$_2$ emissions and satellite SO$_2$ measurements was developed. The method is based on fitting satellite SO$_2$ vertical column densities (VCDs) to a set of functions of OMI pixel coordinates and wind speeds, where each function represents a statistical model of a plume from a single point source. The concept is first demonstrated using sources in North America, and then applied to Europe. The correlation coefficient between OMI-measured VCDs (with a local bias removed) and SO$_2$ VCDs derived here using reported emissions for 1° by 1° gridded data is 0.91 and the best-fit line has a slope near unity, confirming a very good agreement between observed SO$_2$ VCDs and reported emissions. Having demonstrated their consistency, seasonal and annual mean SO$_2$ VCD distributions are calculated, based on reported point-source emissions for the period 1980-2015, as would have been seen by OMI. This consistency is further substantiated as the emissions-derived VCDs also show a high correlation with annual mean SO$_2$ surface concentrations at 50 regional monitoring stations.





## 1 Introduction

Sulfur dioxide ($SO_2$) is a designated criteria air pollutant that enters the atmosphere through anthropogenic (e.g., combustion of sulfur-containing fuels, oil refining processes, metal ore smelting operations) and natural processes (e.g., volcanic eruptions and degassing). Over the past three decades both the US and Canada have taken measures to reduce atmospheric

emissions of $SO_2$ in order to combat acidification of the ecosystem (e.g., acid rain) and fine particulate matter. As a result, between 1990 and 2012, reported emissions of $SO_2$ declined by 78 percent in the United States and 58 percent in Canada (IJC, 2014). In this study, we examined how well the changes in the reported emissions agree with the $SO_2$ changes in North America observed by satellite and surface instruments.

Ground-based networks such as the US Clean Air Status and Trends Network (CASTNet) and Canadian Air and

Precipitation Monitoring Network (CAPMoN) are specifically designed to monitor long-term trends of gaseous pollutants in rural areas away from major pollution emission sources (Baumgardner et al., 1999; Park et al., 2004; Schwede et al., 2011). Their measurements show that over the eastern US, reductions in regional $SO_2$ emissions have led to significant reductions in monitored $SO_2$ concentrations (Sickles II and Shadwick, 2015; Xing et al., 2013).

Satellites provide global measurements of $SO_2$ vertical column densities (VCD): the total number of molecules or

total mass per unit area (Krotkov et al., 2008; Li et al., 2013; Theys et al., 2015). They have been previously used to study the evolution of $SO_2$ VCDs over large regions such as Europe (Krotkov et al., 2016), China (Jiang et al., 2012; Koukouli et al., 2016; Li et al., 2010; Witte et al., 2009), India (Lu et al., 2013), and the U.S. (Fioletov et al., 2011). Satellite instruments can detect anthropogenic $SO_2$ signals from large individual point sources such as copper and nickel smelters, power plants, oil and gas refineries , and other sources (Bauduin et al., 2014, 2016; Carn et al., 2004, 2007; Fioletov et al., 2013; de Foy et

al., 2009; Lee et al., 2009; McLinden et al., 2012, 2014; Nowlan et al., 2011; Thomas et al., 2005). An 11-year-long record of satellite $SO_2$ data over different regions of the globe, including the eastern US and southeastern Canada, was examined recently (Krotkov et al., 2016). The analysis shows a substantial (up to 80%) decline in the observed VCD values over that region.

These satellite measurements can also be used as an independent source to verify reported changes in emissions.

Methods for emission estimates from satellite measurements have been recently reviewed by (Streets et al., 2013). One such method that does not require the use of atmospheric chemistry models has been commonly used in recent years. By first merging observations from the Ozone Monitoring Instrument (OMI) with wind information, the downwind decay of several pollutants can be analyzed, and in so doing estimates of the total $SO_2$ (or $NO_2$) mass ($\alpha$) near the source and its lifetime or, more accurately, decay time ($\tau$) can be derived (Fioletov et al., 2011, 2015; de Foy et al., 2015; Lu et al., 2013, 2015; Wang

et al., 2015). The emission strength ($E$) can be obtained using the expression $E=\alpha/\tau$ if we assume a steady state for these quantities. The mass can be derived directly from satellite measurements, while the lifetime can either be prescribed using known emissions (Fioletov et al., 2013) or estimated from the measurements based on the rate of decay of VCD with distance downwind (Beirle et al., 2014; Carn et al., 2013; de Foy et al., 2015). Model-based comparisons of different



methods to estimate $E$ and $\tau$ demonstrate that such methods can produce accurate estimates of $\tau$ (de Foy et al., 2014). In our previous study (Fioletov et al., 2015), values of $\alpha$ and $\tau$ for anthropogenic point sources were derived from OMI measurements by fitting a 3-dimensional function of the geographic coordinates and wind speed.

These methods, however, are applicable to individual point sources. When this condition is not met, as is the case for
multiple sources, the sources can either be combined together if they are close (Fioletov et al., 2015), or the fitting domain is split and the sources are fit separately (Wang et al., 2015). Both approaches have their limitations. In this study, we derive a general relationship between emissions and VCDs that can be used for the estimation of emissions from multiple sources. Moreover, the approach can be used in reverse: that is, VCDs can be estimated directly from reported emissions data, thus making it possible to study the link between VCDs and surface concentrations even for the period before satellite
measurements became available. This study is focused on the eastern U.S. and southeastern Canada, where the majority of large North American $SO_2$ emissions sources (mainly coal-burning power plants) are located, where the changes in both reported emissions and measured VCDs are particularly large, and where emissions are measured directly at the stack for most sources. In this region, there is also a network with long-term records of uniform $SO_2$ surface concentration measurements. All of this makes it possible to study consistency between the measurements of emissions, VCDs, and surface
concentrations. Once the link between these measurements is verified, it is possible to estimate one measured quantity from another. As an illustration, we demonstrate how European $SO_2$ emissions can be estimated from OMI VCD data.

## 2 Data Sets

### 2.1 Satellite $SO_2$ VCD data

OMI, a Dutch-Finnish UV-Visible wide field of view nadir-viewing spectrometer flying on NASA's Aura spacecraft
(Schoeberl et al., 2006), provides daily global coverage at high spatial resolution (Levelt et al., 2006). OMI has the highest spatial resolution and is the most sensitive to $SO_2$ sources among the satellite instruments of its class (Fioletov et al., 2013). Operational OMI Planetary Boundary Layer (PBL) $SO_2$ data produced with the Principal Component Analysis (PCA) algorithm (Li et al., 2013) for the period 2005-2015 were used in this study. Retrieved $SO_2$ VCD values are given in Dobson Units (DU, 1 DU = $2.69 \cdot 10^{26}$ molec$\cdot$km$^{-2}$).
OMI $SO_2$ VCD data are retrieved for 60 cross-track positions (or rows). In order to use only data with the highest spatial resolution, we excluded data from the first 10 and last 10 cross-track positions from the analysis to limit the across-track pixel width from 24 km to about 40 km, while the along-track pixel length was about 15 km (de Graaf et al., 2016). In other words, a single OMI measurement represents an $SO_2$ VCD value averaged over a 350-500 km$^2$ area.

Measurements with snow on the ground were excluded from the analysis as the OMI PCA algorithm presently does
not account for the effects of snow albedo. Only clear-sky data, defined as having a cloud radiance fraction (across each pixel) less than 20%, and only measurements taken at solar zenith angles less than 70° were used. Beginning in 2007, up to a half of all rows were affected by field-of–view blockage and stray light (the so-called "row anomaly") and those affected





pixels were also excluded. Additional information on the OMI PCA $SO_2$ product can be found in other publications (Krotkov et al., 2016; McLinden et al., 2015).

$SO_2$ VCD data from the Ozone Mapping Profiler Suite (OMPS) Nadir Mapper on board the Suomi National Polar-orbiting Partnership (or Suomi NPP) satellite operated by NASA/NOAA and launched in October 2011 were also used in the

study to verify a potential bias in some OMI data (see the Supplement, section S1). OMPS data were processed with the same PCA algorithm as OMI data (Li et al., 2013; Zhang et al., 2017). OMPS has a lower spatial resolution than OMI, 50 km by 50 km, but better signal-to-noise characteristics.

To eliminate cases of transient volcanic $SO_2$, periods when $SO_2$ values observed over the eastern U.S. were affected by volcanic emissions; we determined and excluded such cases from the analysis. The range of analyzed $SO_2$ VCD values

was limited to a maximum of 3 DU. Since the average $SO_2$ value over the largest $SO_2$ source in the US is about 1 DU and the standard deviation of individual measurements is 0.5 DU, such a limit corresponds to the 4 standard deviations level even over even the largest sources. Of the $SO_2$ values over the eastern U.S. and southern Canada considered here, the years 2008 and 2009 are particularly problematic due to the eruptions of Kasatochi (Aleutian Islands, Alaska, August 2008, 52°N) and Sarychev (Kuril Islands, Eastern Russia, June 2009, 48°N). High volcanic $SO_2$ values were also observed on several days in

2011. In addition to the filtering based on $SO_2$ values, five time intervals were explicitly removed from the analysis to avoid misinterpretation of volcanic $SO_2$ as anthropogenic pollution. The intervals are: 07.07.2008–23.07.2008, 08.08.2008–08.09.2008, 23.03.2009–10.04.2009, 16.06.2009–05.07.2009, and 22.05.2011–09.06.2011. To remove volcanic $SO_2$ in the case of Europe, the analyzed data were divided into 5° by 5° cells, and for each cell, days with the 90[th] percentile above a 5 DU limit were excluded from the analysis. Only about 1.5% of all data were removed by this screening.

**2.2 Wind data**

As in several previous studies (Fioletov et al., 2015; McLinden et al., 2016), wind speed and direction data for each satellite pixel were required for the analysis methods applied. European Centre for Medium-Range Weather Forecasts (ECMWF) reanalysis data (Dee et al., 2011) (http://apps.ecmwf.int/datasets/) were merged with OMI measurements. Wind profiles are available every 6 hours on a 0.75° horizontal grid and are interpolated in time and space to the location of each OMI pixel

center. U- and V- (west-east and south-north, respectively) wind-speed components were first averaged in the vertical between 0 and 1 km where the majority of the $SO_2$ mass resides. The wind components were then interpolated spatially and temporally to the location and overpass time of each OMI pixel.

Note that to reconstruct annual mean VCD maps based on annual emissions (section 3.4), it is not necessary to have the actual year-specific meteorological information, as annual mean wind characteristics do not vary much from year to year,

and so for convenience, we simply used wind data from 2005 for all years prior to 2005.



## 2.3 SO$_2$ emissions inventories

Monthly or annual emissions from individual US point sources available from the U.S. Environmental Protection Agency (EPA) National Emissions Inventory (https://www.epa.gov/air-emissions-inventories) for the period 1980-2015 were examined in this study. U.S. EPA national emissions inventories are available from 1980, although at that time they

contained just annual values and were updated only every 5 years. Regular annual emissions data for consecutive years first became available in 1995 and U.S. emissions data with higher temporal resolution (monthly, daily, and hourly) are only available after 2004. Note that the inventory data for these sources after the early 1990s were based on direct stack measurements by Continuous Emissions Monitoring Systems as mandated by Title IV of the 1990 U.S. Clean Air Act Amendments (Public Law 101-549) (e.g., https://www.epa.gov/clean-air-act-overview). The Canadian SO$_2$ annual point-

source emissions data were obtained from the National Pollutant Release Inventory (NPRI), http://open.canada.ca/data/en/dataset/). Canadian annual point-source emissions data sets are available back to 2002 and we used the 2002 emissions data for the 1980-2001 period. For Canadian sites, only annual emissions are available and seasonal values were calculated by dividing annual emissions by 4. This study is based on point-source emissions only, but point sources have contributed a large majority (>90% in the early 2000s and >70% in the recent years) of North American SO$_2$

emissions.

Information about point source emission from the European Union (EU) countries from the European Pollutant Release and Transfer Register (E-PRTR) for 2004-2014 is available from http://www.eea.europa.eu/data-and-maps/data/lcp-1 and was used for the analysis for Europe. For non-EU European countries, spatially distributed 2005-2014 TNO-MACC-III emission data for air pollutants from the MACC project was used (Kuenen et al., 2014) (Monitoring Atmospheric

Composition and Climate; see http://www.gmes-atmosphere.eu/) prepared by TNO. When E-PRTR data are not available, proxy data are used by TNO, such as for power plants from the World Electric Power Plants Database (WEPP; see http://www.platts.com/products/world-electric-power-plants-database). WEPP provides no emission data, only listing unit characteristics, so emissions are allocated to individual plant units based on the reported thermal capacity, configuration and generic interpretations of reported fuel type(s), and installed emission control technologies. Site-specific parameters not

provided by WEPP, such as exact fuel sulphur content, achieved pollutant removal efficiencies, and load fluctuations, are not taken into account when emissions are allocated. Therefore, the MACC-III point source emission data should be regarded as estimates that may differ considerably from the actual emissions.

## 2.4 SO$_2$ surface concentration data

In-situ SO$_2$ ground-level measurements from the U.S. Clean Air Status and Trend Network (CASTNet) (Baumgardner et al.,

1999; Park et al., 2004; Schwede et al., 2011), operated by the U.S. EPA (http://www.epa.gov/castnet), and the Canadian Air and Precipitation Monitoring Network (CAPMoN: http://www.ec.gc.ca/rs-mn/default.asp?lang=En&n=752CE271-1) (Schwede et al., 2011), operated by Environment and Climate Change Canada (ECCC), were used in this study. Both



networks were established to assess regional trends in pollutant concentrations, atmospheric deposition, and ecological effects due to changes in air pollutant emissions. CASTNet started operations in 1987 and CAPMoN started in the late 1970s. Both networks employ filter packs to measure $SO_2$, although CASTNet uses a one-week sampling period vs. a one-day sampling period for CAPMoN. It is important to note that the monitoring sites belonging to these networks are located

in relatively remote areas, so that direct impacts of local pollution sources on the measurements are minimal. Annual mean $SO_2$ values in µg m$^{-3}$ were used in this study.

## 3 Linking satellite $SO_2$ VCDs and $SO_2$ emissions

The method for linking OMI $SO_2$ VCDs to $SO_2$ emissions is based on a fit of OMI VCDs to an empirical plume model developed to describe the $SO_2$ spatial distribution (as seen by OMI) near emission point sources (Fioletov et al., 2015), but

unlike the previous studies, it is not limited to a single point source. The plume model assumes that the $SO_2$ concentrations emitted from a point source decline exponentially with time and that they are affected by turbulent diffusion that can be described by a 2D Gaussian function. Each satellite measurement (or pixel) is fit by a sum of plumes from all point sources. The distribution of $SO_2$ emanating from each source is described by the plume model based on a known plume function $\Omega$ $(\theta, \varphi, \omega, s, \theta_i, \varphi_i)$ dependent on the satellite pixel coordinates $(\theta, \varphi)$, pixel wind direction and speed $(\omega, s)$, and source

coordinates $(\theta_i, \varphi_i)$ scaled by an unknown parameter $(\alpha_i)$ representing the total $SO_2$ mass from the source $i$. These unknown parameters are then estimated from the best fit of the OMI measurements. The emission rate for source $i$ is $E=\alpha_i/\tau$, where $\tau$ is a prescribed $SO_2$ decay time. In other words, the method finds the emission rates that produce the best agreement with the observed OMI $SO_2$ VCD values. The detailed formulas are given in the Appendix.

       Thus, the fitting procedure allows for the isolation of the emissions-related "signal" in the data from known sources

and can be used to check existing point-source emissions inventories. If all sources are included in the fit, it can be expected that the difference between the OMI data and the fit is within the noise level and the estimated emission rates $E$ should agree with the reported emissions. We used OMI observations and emissions data for the eastern U.S. and southeastern Canada to confirm this expectation. Sources that are not included in the fit would appear as "hotspots" on the maps of the difference between OMI VCDs that could be used for source detection. Furthermore, emissions from such sources could then be

derived by adding their coordinates to the source list in the fitting procedure. The suggested method can thus be used as a source of independent emission estimates in regions where emissions values have large uncertainties.

       The method requires information about the point-source locations. We used source location data available from the US and Canadian emission inventories mentioned in section 2.3. As discussed by (Fioletov et al., 2015), sources that emit 30 kt yr$^{-1}$ or more can be detected by OMI. Since multiple smaller sources located in a close proximity can also be seen as a

"hotspot" in OMI data, we lowered the minimum limit and included all $SO_2$ point sources that reported emissions of 20 kt yr$^{-1}$ or more at least once in the period 2005-2015. It should be noted that while the method does not improve the level of source detectability, it gives more accurate emission estimates for clusters of small sources where the point source algorithm is not really applicable.



Earlier versions of the OMI SO$_2$ data product have some large-scale biases (Fioletov et al., 2011) that were largely removed in the present PCA version. However, we found that even the PCA version has some biases that may interfere with the regression fit. The bias can be accounted for by introducing functions that change slowly with latitude and longitude. We used Legendre polynomials of latitude and longitude and their products that are orthogonal over the analysed domain as discussed in the Appendix.

The OMI data with and without the bias and the fitting results for four multi-year intervals are shown in the two left-hand columns of Figure 1. The additional plots of the bias itself and the residuals are available from the Supplement, Figures S1 and S2. Figure 1 is based on the annual estimates averaged over two- and three-year periods. Figure 1 confirms that there was a large decline in SO$_2$ VCD over the eastern U.S. and southeastern Canada in the period 2005-2015 (Krotkov et al., 2016). In contrast, the bias estimated from the fitting procedure appears to be fairly constant over time (Figure S1), which suggests that it may be an artifact from the retrieval. The lack of this feature in OMPS observations further suggests it is a bias in OMI PCA data as discussed in the Supplement (Section S1 and Figure S2).

It should be mentioned that the use of an empirical plume model is appropriate when atmospheric advection/diffusion can be considered to be the dominant process and meteorological conditions can be assumed to be quasi-steady. This is a reasonable assumption for short time periods and transport distances and when chemical transformation and surface removal of SO$_2$ can be well represented as simple first-order loss. The consistent mid-day overpass time for OMI means that the vast majority of the satellite measurements will be associated with a well-developed quasi-steady planetary boundary layer. A 3D atmospheric chemistry model, on the other hand, would be more appropriate for longer time periods and transport distances and for emissions occurring at all times of day, but that is not the case for this analysis.

## 4 Analysis

### 4.1 SO$_2$ emission estimates from OMI data

The functions $\Omega$ $(\theta, \varphi, \omega, s, \theta_i, \varphi_i)$ decline very rapidly with distance from the source located at $\theta_i, \varphi_i$. For an isolated point source $(\theta_i, \varphi_i)$ where other sources are located 100 km away or more, $\Omega$ $(\theta, \varphi, \omega, s, \theta_i, \varphi_i)$ is not correlated with any other $\Omega$ $(\theta, \varphi, \omega, s, \theta_j, \varphi_{j,})$, where $i \neq j$, and the regression model (1) or (2) can be simply split into two parts: a model for point-source emission estimates for source $i$ and a model for all other point sources. Then the estimate of $\alpha_i$ is independent from estimates for all other sources. If, however, there is another source $j$ located at $(\theta_j, \varphi_j)$ that is closer to source $i$ than ~100 km, then the functions $\Omega$ $(\theta, \varphi, \omega, s, \theta_i, \varphi_i)$ and $\Omega$ $(\theta, \varphi, \omega, s, \theta_j, \varphi_{j,})$ become correlated, as do their estimates of $\alpha_i$ and $\alpha_j$. As the two $\Omega$ functions depend on the wind, the correlation coefficients also depend on the wind distribution and the locations of the sources relative to the prevailing wind direction and to each other, but the separation distance is the dominant factor. Typical absolute values for the correlation coefficients are about 0.2, 0.6, and 0.8 for distances between sources of 100 km, 50 km, and 25 km, respectively. A high correlation means that, in practice, emissions estimates for sources located in close proximity have large uncertainties as we may have difficulty separating signals from the individual sources. However, if





sources $i$ and $j$ are located in close proximity to each other but far from all other sources, then their combined emissions can still be estimated accurately. Thus, such sources can be grouped into clusters, where the member sources are located in close proximity (20-40 km) but the clusters themselves are well-separated and total emissions from each cluster can be estimated from satellite data.

5        Another way of grouping sources into clusters is to establish a grid over the analysis region and then sum up estimated emissions ($E_i$) from all sources within each grid cell. Of course, this does not prevent situations in which two sources are in close proximity but are located in adjacent grid cells. Such cases would lead to larger uncertainties in the cell values, but they are uncommon. Figures 2 b-d show examples of such estimated total emissions for three 1° by 1° cells. Seasonal emissions estimates scaled to annual values were used for this plot and winter data are not shown in this plot due to

much higher uncertainties of OMI data. The estimated emissions agree reasonably well with the emissions calculated by summing up reported $SO_2$ emissions from the point sources in each cell. The standard deviation of the difference between the emission estimates for all 1° by 1° cells within the domain area shown in Figure 1 and reported $SO_2$ emissions for the same cells are 112, 39, 28, and 41 kt yr$^{-1}$ for winter, spring summer, and autumn, respectively. The standard deviations of the difference are 25 kt yr$^{-1}$ and 37 kt yr$^{-1}$ for annual emissions without and with winter data (not shown), respectively.

Finally, total point-source emissions for the entire region can be estimated by summing over all individual point sources. Such a plot is shown in Figure 2a. The estimated $SO_2$ emissions in Figure 2a follow the trend in the reported emissions well, and the correlation coefficient between the two data sets is 0.98. The agreement is particularly good in summer. Large discrepancies are observed only in autumn months after 2007, when relatively high measurement noise combined with the reduction of data due to the row anomaly. In addition, the 2008 and 2009 satellite data were affected by $SO_2$ emitted from

volcanic eruptions (McLinden et al., 2015). More information on the autumn data is available from Sections S2 and S3 of the Supplement.

       This grid-based approach can be potentially used for area sources or when the locations of sources are not well known. For illustration, we used VCD measurements over the same area, but assumed that that it is an area source with no individual point sources. If we set a regular grid and assume that each grid point is a "source", we can estimate emissions

from such "sources" as described above. VCD can then be calculated using these estimated emissions. Such reconstruction for a 0.5° by 0.5° grid is also shown in Figure 1 and demonstrates a good agreement with the measured VCD values. Note that the grid spacing should not be too large or else the areal emissions will be underestimated. Likewise, if it is too fine adjacent grid cells will be highly correlated and may result in artificial structure. As Figure 1 (column 4) shows, the fitting results based on emissions are very close to the OMI fitted data (column 2). We used the OMI data with local bias removed

because with this approach, any instrumental local bias will be interpreted as an area source, resulting in overestimation of emissions.

       Emissions estimated by this gridded method are also shown in Figure 2. Their uncertainties are higher than for the case of known source locations but are still reasonable. The standard deviation of the difference between the emission estimates for all 1° by 1° cells within the domain area shown in Figure 1 and reported $SO_2$ emissions for the same cells are



54, 37, and 56 kt yr$^{-1}$ for spring summer, and autumn, respectively. High measurement errors and data gaps prevent estimation of the emissions for winter.

The uncertainties of satellite-based emission estimates have been discussed in our previous studies (Fioletov et al., 2015, 2016). They can be as high as 50%, but the two largest contributors to this uncertainty, the airmass factor (AMF; determined by the assumed vertical profile, surface reflectivity, and viewing geometry) and the prescribed lifetime, are related to site-specific conditions and can be considered primarily as systematic. They introduce a scaling factor in estimated emissions that affects absolute values but not relative year-to-year changes in emissions. Moreover, the constant, effective AMF embedded in the OMI SO$_2$ product is based on measurements taken in the eastern US, and the lifetime estimates used here are based on data from the US power plants as well, so these errors are minimal for this region. To further support this claim, AMF values were recalculated for all SO$_2$ observations used in Fioletov et al. (2016) and its impact on these sources was found to minimal, typically less than 5%.

## 4.2 SO$_2$ VCDs estimated from reported emissions

The equation that links emissions and VCDs (A1) can also be used for forward calculations: if coefficients $\alpha_i$ are known, then SO$_2$ VCDs can be calculated for any location for given wind conditions and these daily VCDs can be averaged to give annual or seasonal means for the analyzed area. Since $\alpha_i = E_i \cdot \tau$, and $\tau$ is prescribed in our calculations, the available emission inventories that contain $E_i$ can be used to calculate $\alpha_i$. In this case, there is no need to do any fitting or to use any OMI measurements to calculate VCDs. In practice, we can simply use the reported emission data and available OMI pixel locations merged with the wind information and calculate VCDs for each OMI pixel based on its center coordinates. OMI provides daily near-global coverage, and of course, no pixel screening is required for such forward calculations, so it would be essentially a reconstruction of daily VCD maps with spatial resolution of about 15 km by 35 km (approximately the average size of the OMI pixel used in this study) assuming a constant emission rate.

Figure 1 (the "VCD from emissions" column) also shows the result of such annual reconstructions averaged over 2- to 3-year periods. Annual point-source emissions from the EPA and NPRI inventories were used as inputs. The agreement of the reconstructed VCDs with OMI data (with the bias removed) is very good, and the agreement with the OMI data fitting results is truly remarkable. To characterize the overall agreement with the OMI data, fitting results, and reconstructed emissions-based VCDs, a 1° by 1° grid was established and various statistical characteristics were calculated for the gridded data. The standard deviation of the residuals ε for this grid is 0.025 DU, i.e., about 20 times less than the uncertainty of individual OMI measurements. The standard deviation of the difference between the OMI-fitted and the reconstructed emissions-based VCDs is 0.016 DU.

Figure 3 shows the scatter plots between the annual VCDs reconstructed from emissions and the three OMI-based data sets shown in Figure 1 for all years. The correlation between the VCDs reconstructed from emissions with the actual OMI data is 0.75, but it rises to 0.91 after the local bias is removed and to 0.97 after the emission-related signal is extracted from the OMI data by the fitting procedure (the first term of equation A4). Moreover, values of the latter correlation





coefficient are above 0.88 for all seasonal averages (excluding winter) and they are substantially higher than the correlation coefficients with the actual seasonal OMI data. This result could be used to extract an emissions-related $SO_2$ signal from the OMI data when the signal is weak compared to the noise level but the source locations are known. Additional information is available from the Supplement, Section S2, including a figure of the difference between the fitted VCDs and the reconstructed VCDs as well as seasonal and annual statistics.

Figure 1 shows the fitting results in geographical coordinates, i.e., the first term of equation (A4) from the Appendix was calculated for each OMI pixel without any stratification by the wind speed and direction. However, the fitting itself is done in a four-dimensional space where the wind speed and direction are the other two coordinates. To illustrate the fitting results for different wind speeds, Figure 4 shows the original mean OMI $SO_2$ values (with the bias removed) and the fitting results when the data are binned by the wind speed. The calculations were done for three wind-speed bins for the 2005-2007 period when the $SO_2$ emissions were the highest and the measurements were not affected by the "row anomaly". The wind-speed modal value is about 10 km h$^{-1}$, and the first bin represent calm conditions, the second bin contains measurements taken within ±5 km h$^{-1}$ from the modal value, and the last bin corresponds to relatively high wind speeds. As Figure 4 demonstrates the fitting results are able to capture the changes in $SO_2$ distribution at different wind-speed bins. When the wind speed is low, $SO_2$ values are high over the sources, while the plume spreads out over a larger area when the wind speed is high. The figure also shows that $SO_2$ VCD values measured over the sources, or integrated over a small area around the source, are not good proxies for the emissions because they depend on the wind speed.

## 4.3 Applications for other regions

Direct $SO_2$ emissions measurements are not available for many regions of the globe. The described method can be used to verify or even estimate $SO_2$ emissions for other regions. To test this method further, we applied it to the European region using European Pollutant Release and Transfer Register (E-PRTR) and TNO-MACC emission inventory data (see Section 2.3). Figure 5 is similar to Figure 1, but for a part of Europe where the majority of the $SO_2$ sources are located. Sources that emitted more than 10 kt in any year between 2005 and 2014 are shown on the map as black dots. The limit was lowered to 10 kt yr$^{-1}$ from the 20 kt yr$^{-1}$ value used for North America since clusters of small sources are common in Europe. When the coordinates of the sources were included in the fitting procedure there appeared to be some large-scale local biases particularly over Spain and the Balkan region.

Figure 5 also shows a good general agreement between the OMI data and VCDs estimated from emissions. Both show a substantial $SO_2$ VCD decline over most regions, with $SO_2$ values the highest at the beginning of the analysed period (Spain, Romania, Bulgaria, Greece), No major changes are observed by OMI for power plants in Serbia and in Bosnia and Herzegovina, and they are now producing the highest $SO_2$ VCD values over the domain shown. As their emissions are not in the E-PRTR database, TNO-MACC emission inventory data were used instead.

The method produces estimates for individual sources that can be further grouped in different ways. Estimated and reported annual emissions for the period 2005-2014 were grouped by nation for nine countries with large $SO_2$ emissions, as





shown in Figure 6. There is good agreement qualitatively between the reported and estimated emissions. Some differences in absolute values are expected due to possible multiplicative biases in OMI-based estimates (from the airmass factor and potential errors in $\tau$). In some cases, however, a possible deficiency in the reported emissions cannot be ruled out. For example, OMI-based values for Romania show nearly constant emissions up to 2012 and then a 50% drop, whereas the

reported emissions suggest a steady decline between 2005 and 2013. The uncertainty level of the OMI-based emissions is illustrated in Figure 6 by the panel for Hungary: the total emissions from 3 sources there are below the sensitivity of OMI-based estimates. Figure 6 also shows OMI-based and inventory emissions for Serbia and for Bosnia and Herzegovina. Their inventory emission data are available as estimates based on reported thermal capacity, configuration, and generic interpretations of reported fuel type and may not be accurate. OMI-based estimates provide an independent source for their

verification. For example, the inventory estimates for the copper smelter at Bor, Serbia, are about 4.5 kt $y^{-1}$, i.e, well below the OMI sensitivity level. However OMI sees this source clearly and the OMI-based mean emissions estimate for 2005-2016 is about 70 kt $y^{-1}$, a value in line with high $SO_2$ levels observed there (Serbula et al., 2014). See also Figure S4.

Another clear benefit of the satellite-based method of emission estimates is that such estimates are available with almost no delay. At the time of this study (February 2017), we were able to estimate OMI-based emission for the period

including 2016, while the E-PRTR inventory only reached 2014.

### 4.4 Reconstruction of the past VCD distribution

If detailed emission data are available, it is also possible to calculate emissions-based VCD maps using Equation (A3) for years before the launch of OMI. Figure 7 shows the annual mean VCD maps over the eastern U.S. and southeastern Canada reconstructed from the emissions inventories available since 1980. All point sources (shown by black dots) that emitted

more than 1 kt of $SO_2$ in at least one year during the 1980-2015 period were included in the calculations for a total about 380 sources. Note that we slightly expanded the domain area in all directions to include sources that emitted large $SO_2$ amounts prior to the OMI launch. There are two major periods of dramatic changes in $SO_2$ VCD values: first, in the early 1990s, corresponding to the implementation of the U.S. Acid Rain Program (ARP), established under Title IV of the 1990 Clean Air Act (CAA) Amendments (IJC, 2014). Then beginning in 2009-2010 there are further large reductions attributable to the

installation of additional flue-gas desulfurization units (or "scrubbers") at many US power plants to meet stricter emissions limits introduced by the Clean Air Interstate Rule. The overall decline of total $SO_2$ point-source emissions from the domain area shown in Figure 5 between 1980 and 2015 is 86%.

### 4.5 $SO_2$ surface concentrations and VCDs

Multi-year mean surface $SO_2$ concentrations at stations belonging to the CASTNet and CAPMoN networks (see Section 2.4)

were compared to the estimated VCD values. Maps of multi-year mean surface $SO_2$ concentrations at stations belonging to the CASTNet and CAPMoN networks are shown in Figure 8. The color scheme of Figure 8 was chosen to be comparable to that used in Figure 1. The main features of the VCD and surface concentration distributions are very similar. Both sets of



maps portray a strong decline from the 1980s to 2010s with the highest values observed along the Ohio River, where many coal-fired power plants are located. However, the spatial gradients in the VCD distribution appear to be sharper than in the surface concentration distribution and elevated surface concentrations are spread out over larger areas. For example, $SO_2$ VCDs over Virginia were much lower compared to West Virginia, while $SO_2$ surface concentrations were similar.

5   There are 50 network sites within the domain area shown in Figure 7 that have 15 or more years of observations in the period 1980-2015. A scatter plot of annual mean $SO_2$ surface concentration at all of these sites versus emissions-based $SO_2$ VCD values is shown in Figure 9a for all available years. While there is a clear correlation between the two quantities that reflects similar spatial distributions and temporal trends, the correlation coefficient is not very high (0.83). However, correlation coefficients calculated separately for individual measurement sites are higher, ranging between 0.87 and 0.99.

10 This is illustrated in Figure 9b, where a subset of the scatter plot from Figure 9a for eight sites is shown using different colours for each site.

   Figure 9b also shows that the slopes of the regression lines vary from site to site. If we calculate the slope of the individual regression line for each site (it is essentially the surface-concentration-to-VCD ratio) and then multiply the emissions-based VCDs by that ratio, then we obtain a very good correlation as illustrated by Figure 9c (R=0.986 for the eight

15 sites shown in Figures 9b and R=0.983 for all data points). The regression-line y-intercepts have also been analysed. A positive intercept means that the surface concentration could be non-zero even in the absence of any regional point-source emissions. The estimated intercepts are within ±1.5 μg m$^{-3}$ for all sites except one where the intercepts is 3.5 μg m$^{-3}$. The exception is the CASTNet Horton Station site, located in Virginia 18 km east of the Glen Lyn power plant, whose emissions were about 10 kt yr$^{-1}$ in 2008 and 6.5 kt yr$^{-1}$ in 2011. However, its emissions information was largely missing for the period

20 2009-2015 and this affected our VCD calculations.

   The surface-concentration-to-VCD ratio ultimately depends on the shape of the $SO_2$ vertical profile. The shape could be affected by boundary-layer height, site elevation, and perhaps some local conditions. There are, however, some common features in the ratio distribution. As shown in Figure 9d, the ratio is low in areas of high emissions-based VCDs and low in areas where emissions-based VCDs are low. Of course, it is not the mean VCD value itself that affects the ratio, but proximity to emission sources. Figure 9d is based on VCDs derived from emissions, but the same analysis for OMI-

25 measured VCD demonstrates similar results (the Supplement, Section S5) It may be possible to reconstruct surface concentration distribution from VCDs and additional information such as the planetary boundary layer height (Knepp et al., 2015), but such estimates are outside of the scope of this study.

## 5 Summary and discussion

30 Fitting OMI $SO_2$ VCD data by a linear combination of functions, where each function represents the plume from an individual source, makes it possible to estimate emission from these sources or groups of sources. If the location of all sources is known, it is expected that the fitting results and the actual OMI data will agree within the noise level as was found to be the case for the eastern U.S. and southeastern Canada. The same agreement is also observed for this region if the





reported emissions are used to calculate VCDs. This suggests a simple way of interpreting satellite $SO_2$ VCD data: they should agree with VCD estimates based on available emission inventories or the fitting results based on known source locations.

By applying a statistical plume model (developed from satellite $SO_2$ measurements) to U.S. and Canadian annual $SO_2$ point-source emissions inventories, we were able to reconstruct past annual mean VCDs for the period 1980-2015. High correlation coefficients between the reconstructed VCD values and the actual OMI measurements (0.91 for OMI data with local bias removed) for the period 2005-2015 gives us confidence in both data sets. It also demonstrates that the reported changes in $SO_2$ point-source emissions are reflected by OMI measurements for the period 2005-2015. Moreover, the annual surface $SO_2$ concentrations at the CASTNet and CAPMoN sites also show high correlation coefficients (0.87-0.99) with $SO_2$ VCDs reconstructed from reported emissions. All of these comparisons suggest a high degree of consistency between the reported $SO_2$ point-source emissions and measured $SO_2$ values over the entire 1980-2015 period.

The approach described in this study can be used in several ways. The derived emissions can serve as an independent data source for inventory verification (both point source and gridded): by comparing OMI-estimated $SO_2$ emissions with the inventories or by comparing VCDs calculated from emission inventories to the OMI VCD measurements. It can also provide emissions information for regions where there are no other information sources available. Unreported point and area sources can be detected and emissions from them can be estimates by subtracting VCDs calculated from available emission inventories from satellite VCD measurements. While this study is focused on $SO_2$, the methods can be applied to other species with relatively short lifetimes measured from space, particularly to $NO_2$ and $NH_3$.

We have also applied the method to Europe. The results strikingly illustrate the positive impact of EU legislation; the countries where no decreasing trends are observed are non-EU member states surrounded by EU countries with decreasing emissions. In general, the satellite-based results confirm the trends in reported $SO_2$ emissions from EU member states over the period 2004-2014, but some discrepancies were found that deserve further attention. In one case, for example, it seems that reported emissions already take into account certain planned or foreseen measures, but real-world (satellite-observation) estimates suggest that implementation of these measures was delayed by several years. Moreover, although the trend is clearly followed, the absolute emission levels suggested by the OMI $SO_2$ VCD fitting method are sometimes substantially above the reported emission levels for recent years (Figure 6). Whether these differences are due to underreporting or to methodological issues requires further study.

There are certain limitations to the suggested methods. Satellite $SO_2$ VCD data may still contain local biases that will interfere with emissions estimates or will themselves be interpreted as a source. As the OMI and OMPS data show, these biases could be different from instrument to instrument. Moreover, data from the same OMI instrument could have different biases if processed by different algorithms (Fioletov et al., 2016; their Figure 3). Although the biases could be partially removed using, for example, a constant (for a small fitting area) or polynomial (for larger areas) fit, further improvement of retrieval algorithms is required to eliminate the bias problem. The biases could be particularly large over regions of high $SO_2$ VCD values such as the Persian Gulf and China, so the method should be applied there with caution. The method is also



based on the assumption that all $SO_2$ is located near the surface, which determines the wind data used for the fitting. This may not always be the case for very large sources where $SO_2$ can be lifted into the free troposphere. Finally, the plume model itself may not be optimal in some cases.

**6 Data availability**

5    OMI PCA $SO_2$ data used in this study have been publicly released as part of the Aura OMI Sulphur Dioxide Data Product (OMSO2) and can be obtained free of charge from the Goddard Earth Sciences (GES) Data and Information Services Center.

**Acknowledgments.** We acknowledge the NASA Earth Science Division for funding OMI $SO_2$ product development and analysis. The Dutch-Finnish-built OMI instrument is part of the NASA EOS Aura satellite payload. The OMI project is
10    managed by the Netherlands Royal Meteorological Institute (KNMI) and the Netherlands Agency for Aerospace Programs (NIVR). The US Environmental Protection Agency National Emissions Inventory and the Environment and Climate Change Canada National Pollutant Release Inventory provided $SO_2$ point-source emissions data. OMI PCA $SO_2$ retrievals used in this study have been publicly released as part of the OMSO2 product and can be obtained free of charge at the Goddard Earth Sciences (GES) Data and Information Services Center (DISC, http://daac.gsfc.nasa.gov).





## Appendix

This appendix contains a description of the fitting algorithm used to estimate emissions from point and multiple sources. The algorithm for point sources was previously published by Fioletov et al., (2015), but we briefly repeated it for reader's convenience.

### 5    Fitting algorithm, point source

The first step of the fitting algorithm involves a rotation of the location of each OMI pixel about the source such that, after rotation, all have a common wind direction. Then, the method assumes that concentrations of $SO_2$ emitted from a point source decline exponentially (i.e., as $exp(-\lambda t)$) with time ($t$) with a constant "lifetime" (or decay rate) $\tau = 1/\lambda$. In the absence of diffusion and with a constant wind direction and speed ($s$), $SO_2$ is transported downwind (along the -$y$ axis in the chosen coordinate system) with a concentration that declines exponentially with the distance from the source. Since $t = -y/s$, this decay is simply $exp(\lambda y/s)$ or $exp(\lambda_1 y)$ where $\lambda_1 = \lambda/s$. Likewise, if the wind speed is zero, the distribution of $SO_2$ near the source is governed by diffusion and can be described by a two-dimensional Gaussian function of the distance from the source that depends on one parameter $\sigma$. As both exponential decay of the concentration along the $y$ coordinate and diffusion take place, the overall behaviour can be described as a combination of exponential and Gaussian random variables, also known as an exponentially modified Gaussian function. Therefore, the statistical model of the $SO_2$ plume employed near the point source has the form of a Gaussian function $f(x, y)$ multiplied by an exponentially modified Gaussian function $g(y, s)$: $\Omega(x,y,s) = f(x,y) \cdot g(y,s)$, where $x$ and $y$ (in km) are the coordinates of the OMI pixel center across and along the wind direction, respectively, and $s$ (in km h$^{-1}$) is the wind speed at the pixel center. The model depends on two parameters, the decay time ($\tau$), and the plume width ($\sigma$). It should be multiplied by a scaling factor $\alpha$ that is proportional to the emission strength.

Thus, $OMI_{SO_2} = \alpha \cdot \Omega(x, y, s) = a \cdot f(x, y) \cdot g(y, s)$ where:

$$f(x,y) = \frac{1}{\sigma_1 \sqrt{2\pi}} exp\left(-\frac{x^2}{2\sigma_1^2}\right);$$

$$g(y,s) = \frac{\lambda_1}{2} exp\left(\frac{\lambda_1(\lambda_1\sigma^2 + 2y)}{2}\right) \cdot erfc\left(\frac{\lambda_1\sigma^2 + y}{\sqrt{2}\sigma}\right); \tag{A1}$$

$$\sigma_1 = \begin{cases} \sqrt{\sigma^2 - 1.5y}, y < 0 \\ \sigma, y \geq 0 \end{cases};$$

$$\lambda_1 = \lambda/s;$$

and $erfc(x) = \frac{2}{\sqrt{\pi}} \int_x^\infty e^{-t^2} dt$. The Gaussian function $f(x, y)$ represents the distribution across the wind direction line.



The function $g(y, s)$ is essentially a convolution of Gaussian (determined by the plume width $\sigma$) and exponential functions (determined by $\lambda_1$ related to the lifetime) and represents an exponential decay along the $y$ axis smoothed by a Gaussian function: when $\sigma$ is close to 0, then $g(y, s) \approx \lambda_1 \, exp \, (\lambda_1 y)$ where $y \leq 0$. The wind speed $s$ is included in (A1) only through $\lambda_1 = \lambda/s$. Note that $\sigma_1$ was used in $f(x, y)$ instead of $\sigma$. The value of $\sigma_1$ increased with the distance from the source to

reflect an additional spread of the plume.

Parameters $\sigma$, $\lambda$, and $\alpha$, can be derived from the fit of the OMI observations by the function $\alpha \, \Omega(x,y,s)$, i.e., from a nonlinear regression model. However, if the values for $\sigma$ and $\tau = 1/\lambda$ are prescribed, then the remaining value, $\alpha$, can be determined from a simple linear regression model.

Since $\displaystyle\int_{-\infty}^{\infty}\int_{-\infty}^{\infty} f(x, y) \cdot g(y, s)\,dx\,dy = \int_{-\infty}^{\infty} \left( \int_{-\infty}^{\infty} f(x, y)\,dx \right) \cdot g(y, s)\,dy = \int_{-\infty}^{\infty} g(y, s)\,dy = 1$,

the parameter $\alpha$ represents the total observed number of $SO_2$ molecules (or the $SO_2$ mass) near the source. If $OMI_{SO_2}$ is in DU, and $\sigma$ is in km, then $a$ is in $2.69 \cdot 10^{26}$ molec or 0.029 $T(SO_2)$. Furthermore, the emission strength ($E$) can be calculates as $E = \alpha/\tau$ assuming a simple mass balance.

The function $\Omega$ depends on pixels coordinates in the Cartesian coordinate system related to the wind direction with

the center at the analysed source. These coordinates can be derived from pixel latitude ($\theta$) and longitude ($\varphi$), the wind direction ($\omega$), and the source latitude ($\theta_0$) and longitude ($\varphi_0$), i.e., $\Omega(x,y,s) = \Omega\,(\theta, \varphi, \omega, s, \theta_0, \varphi_0)$. As OMI measurements were merged with the wind data, OMI SO2 VCD at each pixel can therefore be interpreted as a four-dimensional function OMI SO2 $(\theta, \varphi, \omega, s)$. The dependence of $\Omega$ on the model parameters $\tau$ and $\sigma$ is rather complex and we can simplify this approach by assuming that $\tau$ and $\sigma$ are identical for all sources in the analysed region and only the parameter $\alpha$ differs from

source to source (see sensitivity analysis in reference (Fioletov et al., 2016)). Values of $\tau$ and $\sigma$ were selected based on previous estimates for point sources in the eastern U.S. (Fioletov et al., 2015) with some seasonal adjustments: $\tau$ values were =5.6, 6.3, 7.7, and 6.3 hours for winter, spring, summer, and autumn respectively. The plume width $\sigma$=18 km is dependent on multiple factors, but mostly on the OMI pixel size.

**Fitting algorithm, multiple sources**

In case of multiple sources with prescribed $\tau$ and $\sigma$, OMI $SO_2$ VCD can be expressed as a sum of contributions $\alpha_i \cdot \Omega_i$ from all individual sources ($i$). If $(x_i, y_i)$ and $(x'_i, y'_i)$ are the pixel's Cartesian coordinates (km) in the system with the origin at the source $i$ before and after the wind rotation respectively, then they can be calculated from the pixel and source latitudes and longitudes as:

$x_i = r \cdot (\varphi - \varphi_i) \cdot \cos(\theta_i);$

$y_i = r \cdot (\theta - \theta_i);$

$x'_i = x_i \cdot \cos(-\omega) + y_i \cdot \sin(-\omega);$





$$y'_i = -x_i \cdot \sin(-\omega) + y_i \cdot \cos(-\omega);$$

where $r$=111.3 km·$\pi$ /180; $\varphi$ and $\theta$ are the pixel longitude and latitude; $\omega$ is the pixel wind direction (0 for north); $\varphi_i$ and $\theta_i$ are the source $i$ longitude and latitude (all in radian).

5    Then, similarly to equation (S1), the contribution $a_i \cdot \Omega_i$ from the source $i$ can be expressed as $\alpha_i \cdot \Omega_i = \alpha_i \cdot f(x'_i, y'_i)$ $\cdot g(y'_i, s)$, where:

$$f(x'_i, y'_i) = \frac{1}{\sigma_1 \sqrt{2\pi}} exp\left(-\frac{x_i'^2}{2\sigma_1^2}\right);$$

$$g(y'_i, s) = \frac{\lambda_1}{2} exp\left(\frac{\lambda_1(\lambda_1\sigma^2 + 2y'_i)}{2}\right) \cdot erfc\left(\frac{\lambda_1\sigma^2 + y'_i}{\sqrt{2}\sigma}\right);$$

$$\sigma_1 = \begin{cases} \sqrt{\sigma^2 - 1.5 y'_i}, \ y'_i < 0 \\ \sigma, \ y'_i \geq 0 \end{cases};$$

$$\lambda_1 = \lambda / s;$$

(A2)

Thus, OMI SO$_2$ VCD can be expressed as a sum of contributions from all individual sources ($i$) plus noise ($\varepsilon$):

$$OMI_{SO_2}(\theta, \varphi, \omega, s) = \sum_i \alpha_i \Omega(\theta, \varphi, \omega, s, \theta_i, \varphi_i) + \varepsilon ,$$

(A3)

where only parameters $a_i$ are unknown. Equation (S3) represents a linear regression model where the unknown parameters $\alpha_i$ can be estimated from the measured variable ($OMI_{SO2}$) at many pixels and known regressors $\Omega$ ($\theta, \varphi, \omega, s, \theta_i, \varphi_i$). Calculations can be done on an annual or seasonal basis (i.e., using all data for a particular year or for a particular season of a year respectively). Emission estimates for shorter time intervals, e.g., monthly emissions, may be possible for large sources,

15    but they appear to be too noisy for the eastern U.S. and southeastern Canada for practical applications.

Earlier versions of the OMI SO$_2$ data product have some large-scale biases (Fioletov et al., 2011) that were largely removed in the present PCA version. However, we found that even the PCA version has some biases that may interfere with the regression fit if equation (S3) is used. If the fit is done for a relatively small area, the bias can be accounted for by adding a parameter $\alpha_0$ to the equation (S3) and estimating it from the fit:

$$OMI_{SO_2}(\theta, \varphi, \omega, s) = \sum_i \alpha_i \Omega(\theta, \varphi, \omega, s, \theta_i, \varphi_i) + \alpha_0 + \varepsilon ,$$

(A3')

For a larger area, for example for the eastern U.S. and southeastern Canada, geographic variations in the bias can be accounted for by introducing functions that change slowly with latitude and longitude. We used Legendre Polynomials ($P_n(x)$) that are orthogonal on the interval from -1 to +1.





To make the polynomials orthogonal on the analyzed domain, the following transformation was applied:

$L_j(\theta) = P_j(2 \cdot (\theta - \theta_{min})/(\theta_{max}-\theta_{min})-1);$

$L_k(\varphi) = P_j(2 \cdot (\varphi - \varphi_{min})/(\varphi_{max}- \varphi_{min})-1);$

where $\varphi_{min}$, $\varphi_{max}$, $\theta_{min}$, and $\theta_{max}$ are latitudes and longitudes that define the domain area. Then $L_j(\theta)$ and $L_k(\varphi)$, and their

products were added to the fit:

$$OMI_{SO_2}(\theta, \varphi, \omega, s) = \sum_i \alpha_i \Omega(\theta, \varphi, \omega, s, \theta_i, \varphi_i) + \sum_{j+k \leq 6} \beta_{j,k} L_j(\theta) L_k(\varphi) + \varepsilon, \quad (A4)$$

where $\alpha_i$ and $\beta_{j,k}$ are the estimated coefficients. The first sum represents the emission-related fitting and the second sum is the large-scale bias. Equation (S4) also represents a linear regression model and the unknown coefficients can be estimated from the available observations.   Polynomials up to the 6th degree were used for each one-year or one-season fit for the selected

domain (the eastern U.S. and southeastern Canada), although a higher (or lower) degree may be more suitable for a larger (smaller) area (see also section S6). Note that the biases are related to retrieval effects such as imperfection of account for the ozone absorption and therefore are not related to $SO_2$ abundances and not affected by the winds.  For this reason, no dependence of the bias on $s$ is considered.

Figure A1 illustrates the method by using $SO_2$ data from 2005-2007 near the Bowen power plant in Georgia, U.S.

There are 13 sources within the ±200 km square area around the Bowen facility. The fitting was done for every year, estimated values $a_i \cdot \Omega(\theta, \varphi, \omega, s, \theta_i, \varphi_i)$ were calculated for each satellite pixel, then summed up to obtain a $SO_2$ VCD value for the fit for that pixel. For Figure S1, the actual OMI data and the fitting results were averaged over the 2005-2007 period and smoothed by the pixel averaging technique with a 30 km radius. The maps of estimated values for individual sources smoothed in the same way are also shown. The map of the residuals, or the difference between the OMI data-based map and

the fitting results is also shown.

**Table A1.** Legendre Polynomials.

| $n$ | $P_n(x)$ |
| --- | --- |
| 0 | 1 |
| 1 | $x$ |
| 2 | $(3x^2-1)/2$ |
| 3 | $(5x^3-3x)/2$ |
| 4 | $(35x^4-30x^2+3)/8$ |
| 5 | $(63x^5-70x^3+15x)/8$ |
| 6 | $(231x^6-315x^4+105x^2-5)/16$ |



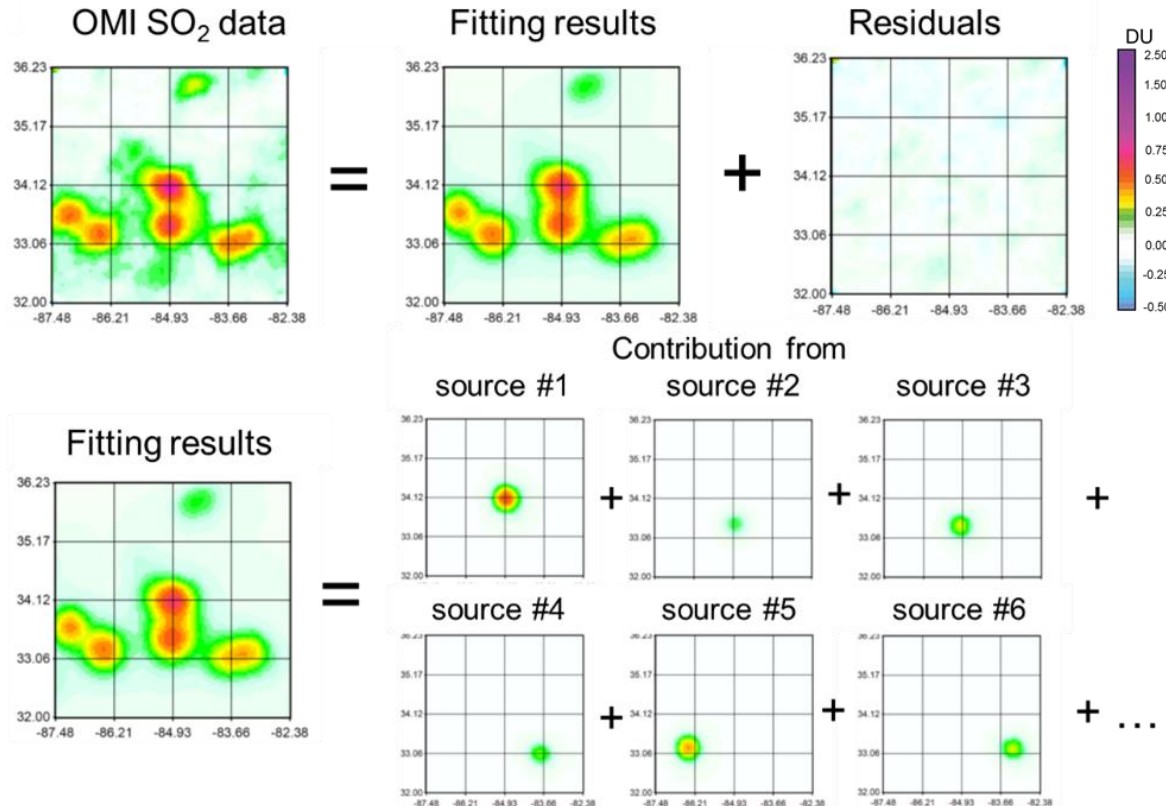

**Figure A1**. Fitting OMI data near the Bowen power plant in Georgia, U.S., 2005-2007. All sources with emissions >20 kt yr$^{-1}$ were included in the fit.



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







**Figure 1**. Annual mean OMI SO$_2$ VCDs from PCA algorithm (column 1), mean OMI SO$_2$ VCDs with a large-scale bias removed (column 2), results of the fitting of OMI data by the set of functions that represent VCDs near emission sources using estimated emissions (see text) (column 3), and SO$_2$ VCDs calculated using the same set of functions but using reported emission values (column 4). Point sources that emitted 20 kt yr$^{-1}$ at least once in the period 2005-2015 were included in the fit (they are shown as the black dots). Results of the fitting of OMI data by the set of functions that represent "sources" as 0.5° by 0.5° grid cells (shown as the black dots) using estimated emissions (see text) are shown in column 5. The maps are smoothed by the pixel averaging technique with a 30 km radius (Fioletov et al., 2011). Averages for four multi-year periods, 2005-2006, 2007-2009, 2010-2012, and 2013-2015, are shown.





**Figure 2**. (a) Reported and estimated seasonal point-source emissions rates (in kt yr$^{-1}$) for the entire eastern U.S. and southeastern Canada (the region shown in Figure 1) for spring, summer, and autumn. (b-d) Examples of reported and estimated seasonal emissions for three 1° by 1° grid cells as labeled on the plots. Estimated emissions are shown for the statistical model based on the actual source location (blue lines) and on a 0.5° by 0.5° regular grid (red lines). Note that the seasonal emissions values are scaled to give annual emission rates. Winter data are not shown due to high uncertainties of OMI measurements.





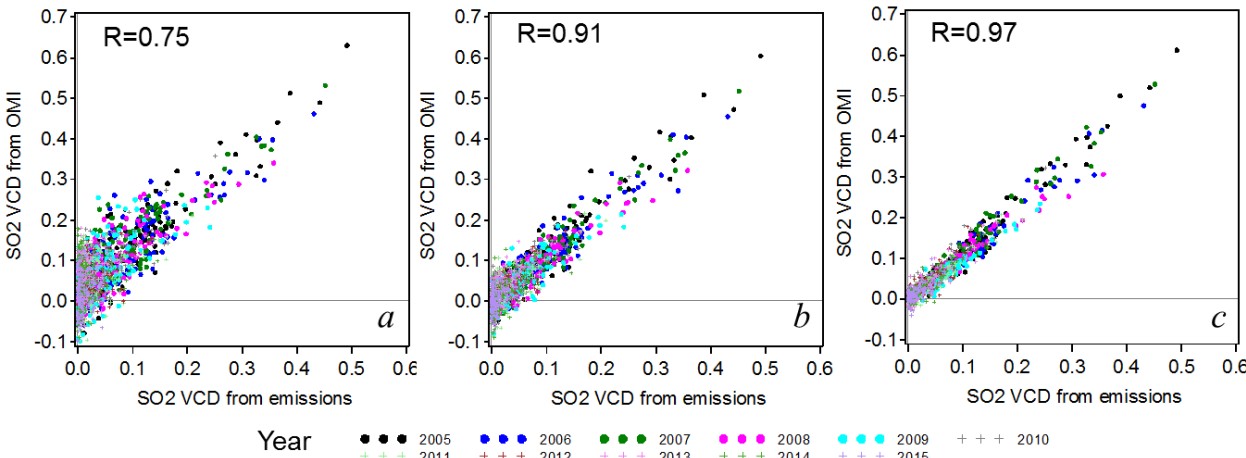

**Figure 3**. The scatter plots between the reconstructed from emissions-based VCDs and the three OMI-based data sets shown in Figure 1: (a) mean OMI SO2 VCDs, (b) mean OMI $SO_2$ VCDs with a large-scale bias removed, and (c) results of the fitting of OMI data by the set of functions that represent VCDs near emission sources using estimated emissions (the first term of equation (A2)). Each symbol on the plot represents the annual mean $SO_2$ VCD value averaged over one 1° by 1° grid cell and all cells within the domain area shown in Figure 1 are included in the plot. Different colours represent different years. The correlation coefficients between the two data sets on each plot are also shown.





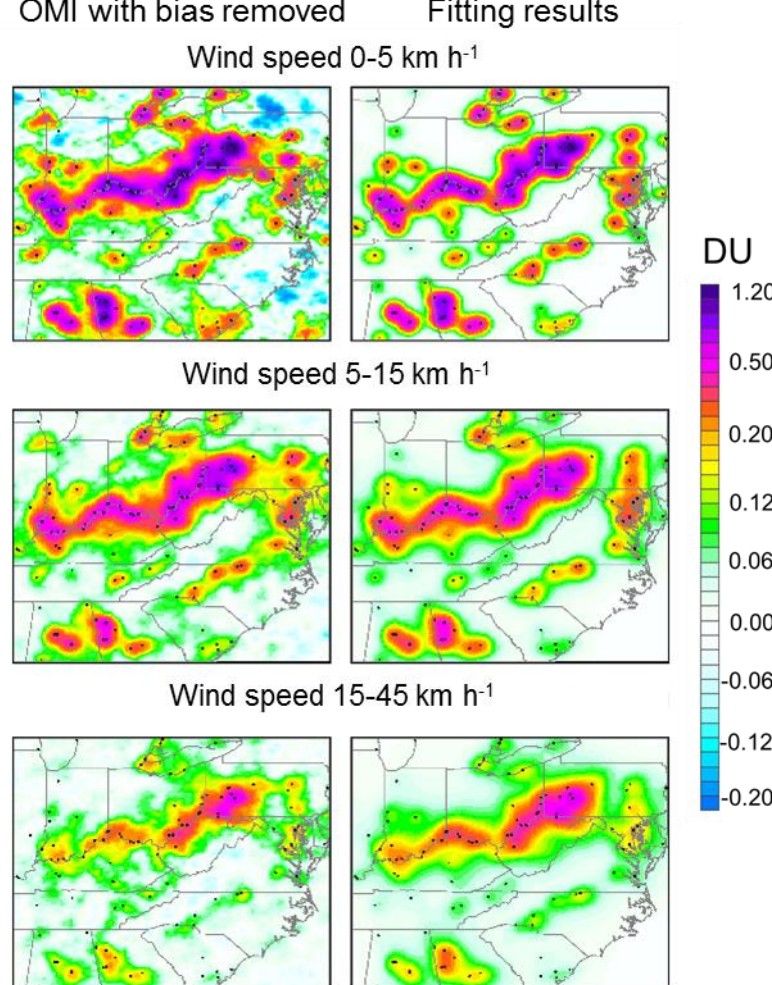

**Figure 4.** (left) Mean OMI SO$_2$ VCDs with a large-scale bias removed and (right) results of the fitting of OMI data by the set of functions that represent VCDs near emission sources using estimated emissions. Averages for 2005-2007 binned by the wind speed (0-5 km h$^{-1}$, 5-15 km h$^{-1}$, and 15-45 km h$^{-1}$) are shown. Sources that emitted 20 kt yr$^{-1}$ at least once in 2005-2015 were included in the fit (they are shown as black dots).



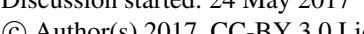


**Figure 5.** The same as Figure 1, columns 1-4, but for the part of Europe where the majority of $SO_2$ point sources are located. Point sources that emitted 10 kt yr$^{-1}$ at least once in the period 2005-2014 were included in the fit (they are shown as the black dots). High $SO_2$ values related to the Mt. Etna volcano in Sicily are excluded from the OMI plots.





**Figure 6.** OMI-based (blue bars) and reported/estimated (black lines) emissions for different European countries. E-PRTR reported emissions were used for all countries except Serbia and Bosnia and Herzegovina, where TNO-MACC estimates (Kuenen et al., 2014) were used (see Supporting Information for details). The error bars represent 2 standard errors of the annual mean calculated by averaging three seasonal (spring, summer, autumn) OMI-based emission estimates





**Figure 7**. Annual mean $SO_2$ VCD calculated using the plume model applied to the reported emissions data. Annual emission data from ~380 $SO_2$ sources (black dots) that emitted 1 kt yr$^{-1}$ at least once in 2005-2015 were included in the calculations.





**Figure 8.** Annual mean surface $SO_2$ concentrations in µg m$^{-3}$ for different periods calculated using data from the CASTNet and CAPMoN surface monitoring networks.







**Figure 9.** (a) A scatter plot of annual mean surface $SO_2$ from CASTNet and CAPMoN vs. VCDs calculated from EPA and NPRI point-source emission inventories. The correlation coefficient between the two data sets is 0.83. (b) A subset of the scatter plot from panel (a) for eight sites (shown by different colors). The correlation coefficients for individual sites are between 0.96 and 0.99. (c) The same plot as (b), but for mean $SO_2$ VCDs multiplied by a site-specific surface-concentration-to-column ratio. The correlation coefficient is 0.98. (d) The site-specific surface-concentration-to-column ratio as a function of the 1980-2015 mean $SO_2$ VCD. Each dot represents one site. Only the 50 regional surface $SO_2$ sites with 15 or more years of data between 1980 and 2015 were used in this figure.