# Peer review of "Multi-source SO2 emissions retrievals and consistency of satellite and surface measurements with reported emissions"

_Atmospheric Chemistry and Physics, 2017_

## Referee Comment (RC1) · Anonymous Referee #1 · 28 Jun 2017

Comments on "Multi-source SO2 emissions retrievals and consistency of satellite and surface measurements with reported emissions" (acp-2017-485) by Fioletov et al.

This paper developed an algorithm to estimate multiple sources SO2 emissions from OMI SO2 VCD. The work is an extension of single SO2 source retrieval from the OMI satellite measurements by the same group. The identification of multiple SO2 emission sources from OMI retrievals has been a challenge. This study moved forward from single source retrieval and made an important contribution to the OMI data applications in a top-down approach to identify and verify the emission sources of criteria and precursor air pollutants. The paper is well-written and publishable in ACP.

[Figure]

I have only several minor questions and comments to the paper as outlined below.

1. pg.7, line 13-15. Does Gaussian point source model take into account atmospheric advection? 2. pg. 7, line 17-18. "a well-developed quasi-steady planetary boundary layer", do you mean a neutral boundary-layer or Ekman layer?

3. pg. 8. line 23. "This grid-based approach can be potentially used for area sources...", Gaussian point source model differs from the area source model. If $SO_2$ emissions derived from Gaussian model, it might not be appropriate to apply Gaussian point source model (Eq. A1) in an area source problem 4. pg 15, line 21, $SO_2$ mass is expressed as 'alpha' after the first equal sign and becomes 'a' after the 2nd equal sign 5. pg 15, line 11-12, ' if the wind speed is zero, the distribution of $SO_2$ near the source is governed by diffusion...'. Diffusion should also depend on the wind and be parameterized by wind. So diffusion should be zero if the wind speed is zero. 6. pg 18., line 9. " Polynomials up to the 6th degree were used for each one-year or one-season fit". Why use the 6th Legendre polynomial? What is difference of retrieved emissions between, say, 6th and 2nd polynomialsïij§

---

## Referee Comment (RC2) · Anonymous Referee #2 · 2 Aug 2017

The study links SO2 emissions as well as surface measurements to column measurements from satellite by a simple dispersion model. While previous studies on this topic focus on individual point sources, this study uses a generalized model function which allows to derive emission estimates for a list of sources (even close to each other) at once. By establishing the link between emissions and columns, even "reconstructed" SO2 columns were generated for the time before actual satellite measurements are available.

The paper is well written. Results are impressive and convincing, and the method is innovative. It should be published in ACP after dealing with the following issues:

[Figure]

General comments:

1. OMI SO2 Bias

The good results are only reached after removing a somehow mysterious "retrieval bias". When reading Page 7, Line 3, I was thinking about some constant, or weakly latitude dependent bias. But in fact, the bias has systematic spatial structure and considerable spatial gradients. The authors argue that the enhanced OMI signal at the East coast is not reflecting true SO2, and in particular the comparison to OMPS is convincing. However, the reasons for this OMI "bias" remain unclear. I don't find the given reasons (O3 interference, stray light) convincing at all.

I see the need for the high degree of Polynomials fitted to remove the unexplained spatial features. However, I would not call it a "bias" (which I would associate with something like a constant offset).

In addition, the authors should

- extend the description of the characteristics of the bias in the paper and point out the spatial pattern (US Eastcoast) in the main text,

- extend the discussion of possible reasons (in paper or supplement),

- be aware that the high degree of the fitted polynomial actually removes any unexpected signal (by adding degrees of freedom, anything can be fitted), thus the good fit results are not that surprising,

- discuss how far the bias removal might affect the emission estimate, in particular for the study on wind dependency (see next point).

2. Dependency on wind speed

The application of the model fit for different wind speed bins is quite interesting. However, the authors do not provide the resulting emission estimates. The authors claim that VCDs are not good proxies for emissions as they depend on wind speed (Page

10, Line 17). But from Figure 4, I have the impression that not only the local, but also the integrated VCD depends on wind speed, which should not be the case according to the model function. Is this the case? Please provide the emission estimates for the 3 wind speed bins. If they are different, discuss possible reasons. Could the difference be related to the fitted Polynomial? Please provide maps of the fitted bias for each wind speed bin in the supplement.

Detailed comments:

Page 4 Line 29: "...do not vary much" - have you checked this? How would the reconstructed VCDs look like if e.g. the wind data from 2006 would be used instead?

Page 6 Line 17: "prescribed SO2 decay time" - please provide details here and give the numbers used for tau for the different seasons.

Page 7 Line 3: "change slowly": this would apply for a polynomial of degree 2, but not for degree 6!

Page 7 Line 11: "artifact from the retrieval": please extend the discussion of the artifact and possible reasons (here or in the supplement).

Page 8: I understand the reason for the structure of Figures 1&2, but the order of the text is a bit confusing: it first refers to Fig. 2b, then Fig. 1, then Fig. 2a, and in the following to particular columns of Fig. 1. Please try to make the text plus references to Figures more smooth. It would also help a lot to have the columns of Fig. 1 labelled (a to e or I to V) to avoid references like 'Figure 1 (the "VCD from emissions" column)'.

Page 8 Line 26: "Figure 1" -> Figure 1 (e)" (or 1 V).

Page 9 Line 32: For the correlation of reconstructed VCDs with OMI (bias removed and emission-related signal extracted), the same model is assumed for both datasets, and any non-matching measurement is removed from the OMI data (by bias removal). Thus, the good correlation is not that surprising.

Page 11 Line 15: "reached *until* 2014".

Page 12 Line 26 end of sentence: dot missing.

Page 13 Line 6: 0.91 is reached after bias removal, as stated in brackets, but these are NOT the "actual OMI measurements" any more!

Page 13 Line 18: I agree in general, but the requirements on spatial resolution and quality of emission inventories would be high, and sources from power stations, industry and traffic are often close to each other. The authors should add a statement that emission inventories with good spatial resolution would be required.

Page 15 Equation A1: The division by wind actually converts the decay rate from time to space reference system. It would be helpful to indicate this by adding a subscript "t" to lambda, and replace "lambda_1" by "lambda_x"

Page 16 Line 4: For y>0, sigma_1 is just sigma, so how far does "sigma_1 increased with the distance from the source"?

Page 16 Line 12: "calculates" -> "calculated"

Figures 1 and 5:

- add lat/lon coordinates.

- add column numbers (a to e or I to V)

Figure 2:

- shift a, b, c to top left corner of panel or even above the panel

- "d" is missing

Figure 8: Why is 1980-1982 included when there have been no measurements?

Figure 9: place labels a-d above panels.

[Figure]

---

## Author Response (AR1)

Comments on "Multi-source SO2 emissions retrievals and consistency of satellite and surface measurements with reported emissions" (acp-2017-485) by Fioletov et al. This paper developed an algorithm to estimate multiple sources SO2 emissions from OMI SO2 VCD. The work is an extension of single SO2 source retrieval from the OMI satellite measurements by the same group. The identification of multiple SO2 emission sources from OMI retrievals has been a challenge. This study moved forward from single source retrieval and made an important contribution to the OMI data applications in a top-down approach to identify and verify the emission sources of criteria and precursor air pollutants. The paper is well-written and publishable in ACP

We would like to thank the reviewer for the evaluation and comments that helped us improve the manuscript.

I have only several minor questions and comments to the paper as outlined below.

1. pg.7, line 13-15. Does Gaussian point source model take into account atmospheric advection?

There is some confusion here. We did use a "pure" Gaussian point source model in our early work, but this study is based on a plume model that combines Gaussian and exponentially modified Gaussian functions as discussed in the Appendix. The latter is responsible for advection. We have added more information about the plume model to the main text.

2. pg. 7, line 17-18. "a well-developed quasi-steady planetary boundary layer", do you mean a neutral boundary-layer or Ekman layer?

We have not assumed any particular boundary-layer type. Depending upon geographic location and time of year, the local boundary layer could be unstable, neutral, or stable. But because the satellite overpass time is close to midday, we do assume that the boundary layer will have adjusted during the morning to any solar heating that occurred.

3. pg. 8. line 23. "This grid-based approach can be potentially used for area sources...", Gaussian point source model differs from the area source model. If SO2 emissions derived from Gaussian model, it might not be appropriate to apply Gaussian point source model (Eq. A1) in an area source problem

We assumed that an area source is a grid of emitting point sources, not just a single point source. Note that our model was developed for plumes as they are seen by the satellite instruments with relatively low spatial resolution.

4. pg 15, line 21, SO2 mass
is expressed as 'alpha' after the first equal sign and becomes 'a' after the 2nd equal sign

Corrected

5. pg 15, line 11-12, ' if the wind speed is zero, the distribution of SO2 near the source is governed by diffusion...'. Diffusion should also depend on the wind and be parameterized by wind. So diffusion should be zero if the wind speed is zero.

Molecular diffusion is always present at any wind speed, and the atmospheric turbulence driving turbulent diffusion can be generated both by mechanical processes and by convective heating, for which the near-midday satellite overpass time is favorable. Moreover, there is always some random error in the wind speed in direction that would also affect $SO_2$ distribution near the source. We changed the text to "…by diffusion or, more generally, random fluctuations..."

6. pg 18.,
line 9. " Polynomials up to the 6th degree were used for each one-year or one-season fit". Why use the 6th Legendre polynomial? What is difference of retrieved emissions between, say, 6th and 2nd polynomials

The problem is that we see some artificially biased $SO_2$ values over some regions. If the area is small, say a few hundred km by a few hundred km, we can simply assume a constant bias. However, for large areas, this assumption does not work and we instead add a function that changes relatively slow with latitude and longitude. The required polynomial degree depends on the area size and the gradients of that slowly changing bias.

This issue was discussed in the Supplement (Section S2):

"The correlation coefficient between OMI data with bias removed and VCDs calculated from the emission data is 0.75 for the actual OMI data, and 0.80, 0.83, 0.87, 0.89, 0.90, 0.909 for the bias removed by the 1st, 2d, 3d, 4th, 5th, and 6th degree polynomials respectively. The correlation noticeably improved if the polynomial bias removed, but the improvement is only marginal for the degrees above 3."

We have added three more figures to the supplement. They show the estimated bias for different polynomial degrees, the fitting results and the emission estimates for $2^{nd}$, $4^{th}$, and $6^{th}$ polynomials. See also our response to the first comment from Reviewer #2.

The study links SO2 emissions as well as surface measurements to column measurements from satellite by a simple dispersion model. While previous studies on this topic focus on individual point sources, this study uses a generalized model function which allows to derive emission estimates for a list of sources (even close to each other) at once. By establishing the link between emissions and columns, even "reconstructed" SO2 columns were generated for the time before actual satellite measurements are
available. The paper is well written. Results are impressive and convincing, and the method is
innovative. It should be published in ACP after dealing with the following issues:

We would like to thank the reviewer for the evaluation and comments that helped us improve the manuscript.

General comments:

1. OMI SO2 Bias

The good results are only reached after removing a somehow mysterious "retrieval bias". When reading Page 7, Line 3, I was thinking about some constant, or weakly latitude dependent bias. But in fact, the bias has systematic spatial structure and considerable spatial gradients. The authors argue that the enhanced OMI signal at the East coast is not reflecting true SO2, and in particular the comparison to OMPS is convincing. However, the reasons for this OMI "bias" remain unclear. I don't find the
given reasons (O3 interference, stray light) convincing at all.

I see the need for the high degree of Polynomials fitted to remove the unexplained spatial features. However, I would not call it a "bias" (which I would associate with something like a constant offset).
In addition, the authors should - extend the description of the characteristics of the bias in the paper and point out the spatial pattern (US Eastcoast) in the main text,
- extend the discussion of possible reasons (in paper or supplement),
- be aware that the high degree of the fitted polynomial actually removes any unexpected
signal (by adding degrees of freedom, anything can be fitted), thus the good fit results are not that surprising,
- discuss how far the bias removal might affect the emission estimate, in particular for the study on wind dependency (see next point).

Perhaps the importance of the polynomial-based bias is somewhat overstated in the paper. While we believe the bias is real, its magnitude is typically within +/-0.1 DU and its impact on emission estimates is rather small (unless we are dealing with area emissions from large areas). We have added about a page of text and three figures to the Supplement that illustrate how the degree of the polynomials affects the bias, the fitting results, and the emissions themselves.
One of the factors that contributes to the bias is surface reflectivity and we have added some discussion about it to the Supplement.

Degrees of freedom is not a big issue here since we are dealing with hundreds of thousands of satellite pixels.

We prefer to use the term "bias" as we used it in our previous work related to point sources where it was indeed a constant offset. To address the reviewer's concern, we have highlighted in a few places that we are dealing with a local bias that that changes relatively slowly with latitude and longitude (compared to signal from emission sources).

2. Dependency on wind speed

The application of the model fit for different wind speed bins is quite interesting. However, the authors do not provide the resulting emission estimates. The authors claim that VCDs are not good proxies for emissions as they depend on wind speed (Page 10, Line 17). But from Figure 4, I have the impression that not only the local, but also the integrated VCD depends on wind speed, which should not be the case according to the model function. Is this the case? Please provide the emission estimates for the

3 wind speed bins. If they are different, discuss possible reasons. Could the difference be related to the fitted Polynomial? Please provide maps of the fitted bias for each wind speed bin in the supplement.

There is some confusion here. We have not estimated the emissions for three wind speed bins. The emission estimates were done using the entire data set. The purpose of Figure 4 is to show that the signal from the same sources would appear differently in OMI data for different wind speeds. OMI SO$_2$ values over the same sources would be higher if the wind speed is low and lower if the wind speed is high. Furthermore, the right column of Figure 4 is not related to actual OMI measurements. It is a reconstruction of VCD distribution based on emission inventories, the plume model, and the actual wind data. We show it to illustrate that we are able to capture the dependence of the OMI SO2 "signal" on the wind speed.

We modified the text to make this clear. As we believe that the bias is related to the retrieval procedure, we assumed that it did not depend on the wind speed. So, the bias is the same for all three wind speed bins.

Estimates for different wind speed bins were discussed in the Supplement (section 5) to our previous paper (Fioletov et al., GRL, 2015)

Detailed comments:

Page 4 Line 29: "...do not vary much" - have you checked this? How would the reconstructed VCDs look like if e.g. the wind data from 2006 would be used instead?

We added such a figure to the Supplement.

Page 6 Line 17: "prescribed SO2 decay time" - please provide details here and give the numbers used for tau for the different seasons.

Details are given in the Appendix. We have changed the text to "The detailed formulas *and prescribed seasonal decay times* are given in the Appendix" to highlight that.

Page 7 Line 3: "change slowly": this would apply for a polynomial of degree 2, but not for degree 6!

We have added the clarification "change slowly (compared to signal from emission sources)". Yes, degree 6 is high, but the analyzed area is huge.

Page 7 Line 11: "artifact from the retrieval": please extend the discussion of the artifact and possible reasons (here or in the supplement).

50    We added some discussion to the Supplement

Page 8: I understand the reason for the structure of Figures 1&2, but the order of the text is a bit confusing: it first refers to Fig. 2b, then Fig. 1, then Fig. 2a, and in the following to particular columns of Fig. 1. Please try to make the

text plus references to Figures more smooth. It would also help a lot to have the columns of Fig. 1 labelled (a to e or I to V) to avoid references like 'Figure 1 (the "VCD from emissions" column)'.

Corrected as suggested.

Page 8 Line 26: "Figure 1" -> Figure 1 (e)" (or 1 V).

Corrected

Page 9 Line 32: For the correlation of reconstructed VCDs with OMI (bias removed and emission-related signal extracted), the same model is assumed for both datasets, and any non-matching measurement is removed from the OMI data (by bias removal). Thus, the good correlation is not that surprising.

Page 11 Line 15: "reached *until* 2014".

Corrected

Page 12 Line 26 end of sentence: dot missing.

Corrected

Page 13 Line 6: 0.91 is reached after bias removal, as stated in brackets, but these are NOT the "actual OMI measurements" any more!

Changed to "the reconstructed VCDs and the OMI-based values"

Page 13 Line 18: I agree in general, but the requirements on spatial resolution and quality of emission inventories would be high, and sources from power stations, industry and traffic are often close to each other. The authors should add a statement that emission inventories with good spatial resolution would be required.

Corrected as suggested

Page 15 Equation A1: The division by wind actually converts the decay rate from time to space reference system. It would be helpful to indicate this by adding a subscript "t" to lambda, and replace "lambda_1" by "lambda_x"

This may create some confusion: we use "y" as the up/downwind coordinate, so it is more logical to use lambda_y. We also prefer to use the same symbols as in the previous publications.

Page 16 Line 4: For y>0, sigma_1 is just sigma, so how far does "sigma_1 increased with the distance from the source"?

In chosen coordinate system, "y" is negative downwind. We have added a reminder of this point in the text.

Page 16 Line 12: "calculates" -> "calculated"

Corrected

Figures 1 and 5:
- add lat/lon coordinates.
- add column numbers (a to e or I to V)

We added the column numbers (I to V) and specified the lat/lon coordinates of the area in the caption

Figure 2:
- shift a, b, c to top left corner of panel or even above the panel
- "d" is missing

5  Corrected

Figure 8: Why is 1980-1982 included when there have been no measurements?

Corrected. The 1980-1982 panel has been removed.
10
Figure 9: place labels a-d above panels.

Corrected as suggested

[revised manuscript text omitted]

The reason for the bias is not well understood.   However, the difference between OMI and OMPS data, both processed with the PCA algorithm, suggests –that at least some of it̶the bias is of instrumental –origin. Incorrect retrieval assumptions could also be contributors.  I̶Two possibilities here are surface reflectivity and assumed ozone profiles. Since this bias has a seasonal-dependence, a third possibility is a small mismatch between the measurement

5     conditions and the conditions under which the principle spectral components were derived.

If the analyzed area is small, say a few hundred km by a few hundred km, we can simply assume a constant bias. However, for large areas, this assumption does not work and we instead add a polynomial function that changes relatively slowly with latitude and longitude.  The required polynomial degree depends on the area size and the

10    gradients of that slowly changing bias.

The bias estimated using polynomials of 0, 2, 4, and 6th degree is shown in Figure S3. Note that the bias is estimated for every season separately and then the average bias over the entire period was calculated. Due to different sampling, the overall bias is not constant even if a constant (0-degree polynomial) is used.  While there is a

15    noticeable difference between the biases estimated using a constant function or 2$^{nd}$-degree polynomial and the bias based on a 4$^{th}$-degree polynomial, further increases of the polynomial degree do not change the results very much.

[Figure]

**Figure S3.** The large-scale biases estimated using polynomials of 0, 2, 4, and 6th degree.

[Figure]

**Figure S4**. Results of the fitting of OMI data by the set of functions that represent VCDs near emission sources using estimated emissions (the same as column III in Figure 1) for polynomials of 0, 2, 4, and 6th degree used to fit the bias.

[Figure]

**Figure S5**. Reported and estimated seasonal point-source emissions rates for the entire eastern U.S. and southeastern Canada (the region shown in Figure 1) for spring, summer, and autumn for different degrees of the polynomial fit. Estimated emissions are shown for the statistical model based on the actual source location and different colors represent different polynomial degrees of the fitting function. As in Figure 2, the seasonal emissions values are scaled to give annual emission rates. Winter data are not shown due to high uncertainties of the OMI measurements.

The fitting results themselves are shown in Figure S4. The fitting results, i.e., $SO_2$ VCDs derived from estimated emissions using the actual wind data, show little dependence on polynomial degree, suggesting that the emission estimates themselves are not very sensitive to the degree of the polynomial fit.

Figure S5 (similar to Figure 2d) shows the estimated emissions themselves for the entire eastern U.S. and southeastern Canada (the region shown in Figure 1 and in Figures S1-S4) for spring, summer, and autumn. The emission estimates for the entire region are very similar for all presented polynomial fits, i.e., the degree of the fit does not play a major role, at least, in that particular region.

[revised manuscript text omitted]

**S6. Reconstructing the past VCDs using different wind data.**

As mentioned in section 2.2, the actual OMI pixel locations and wind data for 2005 were used to reconstruct annual mean VCD maps based on annual reported emissions (section 3.4) for all years prior to 2005. To study the sensitivity of the results to a particular year (2005) of wind data, we repeated the calculations using 2006 wind data. Figure S9 is similar to Figure 7 and shows annual mean SO$_2$ VCD calculated using the reported emissions data and pixel locations and wind data for 2005 (top) and 2006 (bottom). Annual mean wind characteristics do not vary much from year to year and the reconstruction results for 2005 and 2006 winds are nearly identical.

[Figure]

**Figure S9.** Annual mean SO$_2$ VCD calculated using the plume model applied to the reported emissions data for the period 1980-2003 calculated using the reported emissions data and pixel locations and wind data for 2005 (top) and 2006 (bottom).